# Advances of Reverse Vaccinology for mRNA Vaccine Design against SARS-CoV-2: A Review of Methods and Tools

**DOI:** 10.3390/v15102130

**Published:** 2023-10-21

**Authors:** Maria Karolaynne da Silva, Daniel Melo de Oliveira Campos, Shopnil Akash, Shahina Akter, Leow Chiuan Yee, Umberto Laino Fulco, Jonas Ivan Nobre Oliveira

**Affiliations:** 1Department of Biophysics and Pharmacology, Bioscience Center, Federal University of Rio Grande do Norte, Natal 59064-741, RN, Brazildanielmelo.biomed@gmail.com (D.M.d.O.C.); 2Department of Pharmacy, Daffodil International University, Sukrabad, Dhaka 1207, Bangladesh; shopnil.ph@gmail.com; 3Bangladesh Council of Scientific & Industrial Research (BCSIR), Dhaka 1205, Bangladesh; shupty2010@gmail.com; 4Institute for Research in Molecular Medicine, Universiti Sains Malaysia, Kota Bharu 11800, Kelantan, Malaysia; yee.leow@usm.my

**Keywords:** SARS-CoV-2, mRNA vaccine, reverse vaccinology

## Abstract

mRNA vaccines are a new class of vaccine that can induce potent and specific immune responses against various pathogens. However, the design of mRNA vaccines requires the identification and optimization of suitable antigens, which can be challenging and time consuming. Reverse vaccinology is a computational approach that can accelerate the discovery and development of mRNA vaccines by using genomic and proteomic data of the target pathogen. In this article, we review the advances of reverse vaccinology for mRNA vaccine design against SARS-CoV-2, the causative agent of COVID-19. We describe the steps of reverse vaccinology and compare the in silico tools used by different studies to design mRNA vaccines against SARS-CoV-2. We also discuss the challenges and limitations of reverse vaccinology and suggest future directions for its improvement. We conclude that reverse vaccinology is a promising and powerful approach to designing mRNA vaccines against SARS-CoV-2 and other emerging pathogens.

## 1. Introduction

The severe acute respiratory syndrome coronavirus 2 (SARS-CoV-2) caused a global health and socioeconomic emergency as a result of the unprecedented coronavirus disease 2019 (COVID-19) pandemic. The virus emerged in China in late 2019 and quickly spread across the globe, infecting over 770 million people and killing over 6 million by September 2023 [1]. The COVID-19 pandemic has challenged public health systems and disrupted social and economic activities worldwide, especially with the appearance of new variants that are more contagious and resilient [2,3,4]. Therefore, the pandemic has created an urgent need to develop safe and effective vaccines against SARS-CoV-2 to prevent severe outcomes of the disease and enable social and economic recovery. Vaccines are the best way to protect people and reduce virus transmission, as they stimulate the immune system to recognize and fight SARS-CoV-2 [3,4].

Vaccines are biologics that elicit a specific immune response against a pathogen, providing protection to the vaccinated person. Conventional vaccines rely on the use of either the whole weakened or killed pathogen, or parts thereof (such as proteins or polysaccharides) isolated or recombinant [1,5]. These vaccines have proven effective and safe for many infectious diseases, but they have some drawbacks, such as requiring cultivation of the pathogen under appropriate conditions, posing a risk of reactivation or contamination of the final product, having low immunogenicity for some antigens, and being unable to adapt quickly to new emerging pathogens [6,7].

Introduced in 1989 as a novel class of therapeutic agents, messenger RNA (mRNA) vaccines work by encoding specific antigens derived from an mRNA sequence [8]. Once administered, these mRNAs instruct cells to synthesize the desired proteins within the cytoplasm. These proteins are then displayed on the cell’s surface, initiating immune responses. This process involves antigen-presenting cells (APCs) or the production of antibodies/immunoglobulins, ultimately conferring immunity against particular diseases [6,7].

Reverse vaccinology (RV) is an approach that relies on computational tools to analyze a region of genome, usually encoding a protein, that can generate an immune response to a pathogen in order to identify potential vaccine candidates [9]. These computational tools serve to anticipate antigens that are likely to induce protective responses, as well as the precise regions of antigens, epitopes, recognized by the immune system [10]. Thus, RV allows researchers to identify potential vaccine targets more quickly and efficiently; furthermore, it can be particularly useful for pathogens that are difficult to grow in the laboratory.

In this review, we offer a comprehensive and up-to-date perspective on the use of reverse vaccinology in the design of mRNA vaccines against SARS-CoV-2. Specifically, we emphasize the critical role of computational tools and algorithmic complexities that played an instrumental role in the accelerated development of vaccines for COVID-19. We delve deeply into the steps of reverse vaccinology and compare the in silico tools employed in pertinent studies. Additionally, we summarize and reference significant works that have provided valuable insights into the applications of bioinformatics in addressing the biological challenges posed by COVID-19.

## 2. The Advent of Reverse Vaccinology

RV is a broad term used to describe an approach that employs computational tools to analyze a pathogen’s proteome, identifying potential vaccine candidates [9,10,11]. The term “reverse” in vaccinology highlights the innovative approach of initiating vaccine discovery using computer-analyzed genomic data instead of a live organism, selectively targeting proteins that could serve as potential antigens [12]. This approach took flight in the late 1990s following the complete genome sequencing of the bacterium *Neisseria meningitidis*.

This breakthrough led to the creation and approval of Bexsero, the first-ever vaccine against B strains of *Meningococcal meningitis* (meningococci). Building on this momentum, RV has since been employed to identify antigen candidates for a slew of pathogens, including the hepatitis C virus, influenza, and Zika [5,10]. Furthermore, numerous studies are currently being conducted in experimental trials to investigate the potential of bioinformatics platform-tested multi-epitope vaccines [13,14,15,16,17]. Notably, the principles of RV have also paved the way for the development of mRNA vaccines, showcasing the adaptability and potential of this approach in modern vaccine design.

RV offers a modern approach to vaccine design, presenting several advantages over traditional methods, such as (i) being a more economical choice, reducing the financial burden associated with conventional drug design; (ii) it streamlines the drug design process, cutting down the time traditionally required; (iii) it narrows down the number of proteins under study, ensuring a more focused approach; (iv) it can detect antigens that are present in minute quantities or those expressed during specific phases of an organism’s life cycle; and (v) it is especially beneficial for researching pathogens that cannot be grown using in vitro methods [18,19].

## 3. mRNA Vaccines: A New Era in Immunization

The mRNA vaccines, which include both conventional and self-amplifying mRNA forms, represent a groundbreaking alternative to conventional vaccines. Conventional mRNA-based vaccines encode only the antigen of interest and are characterized by 5${}^{\prime }$ and 3${}^{\prime }$ untranslated regions (UTRs). On the other hand, self-amplifying RNAs not only encode the antigen but also the viral replication machinery, facilitating intracellular RNA amplification and abundant protein expression. Upon entering the host cell, the mRNA undergoes processing and translation, culminating in protein synthesis [20]. These proteins are subsequently presented to the immune system, as depicted in Figure 1. This sophisticated mechanism assists the immune system in recognizing and battling the pathogen. It offers several advantages over traditional vaccines, such as (i) rapid and scalable production; (ii) the ability to modulate antigen expression and stability; (iii) potent activation of both humoral and cellular immune responses; and (iv) the flexibility to quickly adapt to new pathogen variants [6,20,21,22].

Building on this technology, the genetic sequence of the S protein from the SARS-CoV-2 virus serves as a blueprint for the design of mRNA vaccines. These vaccines express either the entire protein or specific segments, such as the receptor-binding domain (RBD), the S1 and S2 subunits, or a sequence of epitopes (Figure 1). The latter approach is not a common design for mRNA vaccines, and represents a novel method of presenting antigens, showcasing the flexibility of mRNA vaccines [6,11,21,23]. The development and distribution of mRNA vaccines comes with challenges. These include the necessity for cold storage and transportation, potential toxicity from lipid nanoparticles, suboptimal mRNA delivery into cells, and concerns regarding adverse reactions or potential autoimmunity [6,7,20].

**Figure 1 viruses-15-02130-f001:**
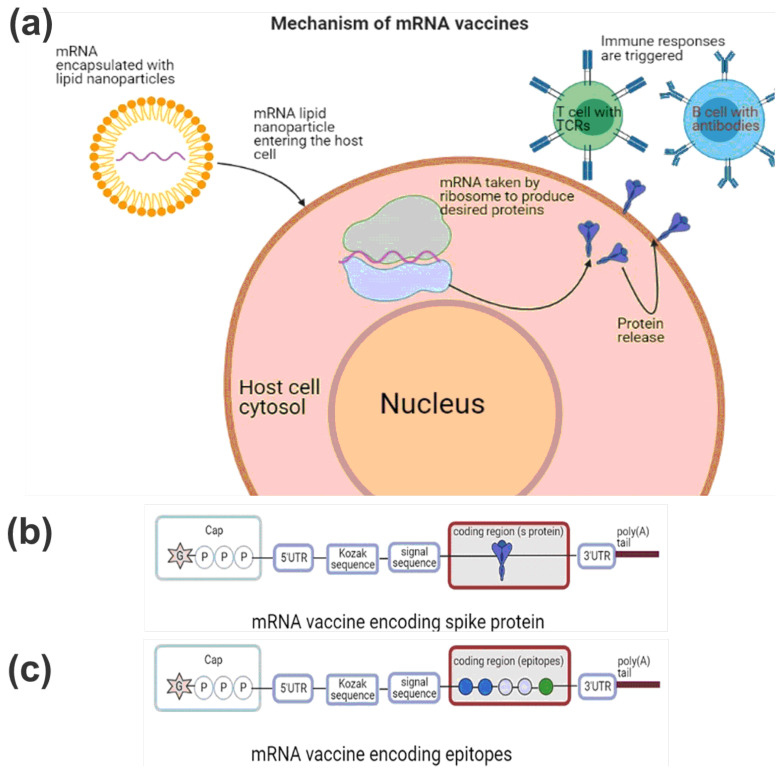
This illustration showcases the mechanism by which mRNA vaccines trigger immune responses. (**a**) The mRNA vaccines elicit an immune response through a multi-step mechanism. Upon administration, the mRNA molecules are taken up by antigen-presenting cells (APCs) at the injection site, such as dendritic cells. The mRNA is then translated by the cellular machinery, leading to the expression of the viral antigen on the surface of the APCs. (**b**) The design of most mRNA vaccines for SARS-CoV-2, notably the BNT162B2, revolves around encoding the spike protein. This crucial protein is located within the open reading frame (ORF) and is entirely encapsulated by LNP. (**c**) A less traditional approach in mRNA vaccine design involves directly encoding epitopes. Unlike the conventional method that incorporates genes or proteins, this design embeds a sequence of epitopes within the ORF. This pioneering method of antigen presentation highlights the adaptability and versatility of mRNA vaccines. Adapted from Cai et al. [24].

### mRNA Vaccines Currently Designed against SARS-CoV-2

Numerous mRNA vaccines have been developed in response to the COVID-19 pandemic, and several have received global approval due to their commendable safety profiles and demonstrated effectiveness in clinical trials. Leading the charge, mRNA vaccines targeting the S protein of SARS-CoV-2 were among the pioneers in clinical testing and quickly secured emergency use approvals in various nations.

Prominent among these are BNT162b2 (Pfizer-BioNTech) and mRNA-1273 (Moderna), both of which advanced swiftly to phase 3 clinical trials. These vaccines reported an impressive efficacy rate of around 95% [25,26]. ARCoV (Walvax Biotechnology) is an mRNA vaccine that remains stable at room temperature for at least a week, underscoring its advantages in distribution and storage [21]. On the other hand, CVnCoV (CureVac) recorded a modest efficacy of 48% against COVID-19.

## 4. Overview of Reverse Vaccinology in mRNA Vaccine Design

While the instability and potential immunogenicity of mRNA initially raised concerns about its use in vaccine development, advancements in research and technology have addressed these challenges. Consequently, established techniques, such as reverse vaccinology, integrated with bioinformatics, have stabilized and streamlined the production process of mRNA vaccines. In relation to SARS-CoV-2, this approach holds promise in identifying conserved and immunogenic epitopes, which is essential for the creation of next-generation multi-epitope or subunit vaccines. The evolving landscape underscores the significance of advanced computational models that effectively integrate epitope data, optimizing vaccine formulations and broadening their applicability.

By focusing on encoding only the vital epitopes in mRNA vaccines, their design becomes more efficient, which in turn improves the stability of the mRNA vaccine. In the context of COVID-19, various studies, as cited by Ahammad and Lira [27], ELKHOLY [28], Bhattacharya et al. [29], Oluwagbemi et al. [30], Khan et al. [31], Oladipo et al. [32], have advocated for a reverse vaccinology approach. This technique starts with obtaining the SARS-CoV-2 spike glycoprotein sequence, and then uses computational predictions to pinpoint epitopes for cytotoxic T lymphocytes, helper T lymphocytes, and linear B lymphocytes, while also evaluating epitope antigenicity.

Addressing the growing concern of SARS-CoV-2 variants, Hussain et al. [33] unveiled DOW-21, a restructured mRNA vaccine construct. This design emphasizes the N-terminal domain (NTD) and the receptor-binding domain (RBD) of the spike protein, integrating variations from both variants of concern (VOCs) and variants of interest (VOIs). The innovative structure merges hypothetical versions of NTD and RBD, combined with a 10-mer gly–ala repeat, and is surrounded by regulatory sequences to enhance intracellular transport and expression. The resulting protein reflects the structural characteristics of SARS-CoV-2 immune escape variants, with the nucleotide sequence optimized for translation efficiency.

Similarly, Durojaye et al. [34] employed a reverse vaccinology approach for an mRNA-based vaccine candidate, focusing on the “YLQPRTFLL” peptide sequence (position 269–277). Identified as a potential B cell epitope with a high affinity for HLA*A-0201, this sequence was selected post antigenicity assessments. The nucleotide design was then customized for the human toll-like receptor 7 (TLR7) after codon optimization.

Beyond the strategies using reverse vaccinology that target the SARS-CoV-2 spike glycoprotein, Pourseif et al. [35] took a unique route. They explored two mRNA vaccine formulations: the domain-based protein vaccine construct (DPVC) and the self-amplifying mRNA vaccine (SAMV). The effectiveness of these designs was confirmed through in silico analyses, which predicted B cell epitopes using various algorithms. They also investigated MHC class I- and II-associated peptide binders, considering factors like allergenicity, autoimmunity, and physicochemical properties. Molecular docking between the vaccine and TLRs 4 and 5 was conducted, and the stability of these complexes was further evaluated using molecular dynamics simulations.

While these studies have yet to be conducted on animals or humans, they provide a compelling demonstration of the potential of computational tools in the development of mRNA vaccines. One of the most promising applications of reverse vaccinology in mRNA vaccine design is the identification of neoantigens [36,37]. Tumor cells produce these novel antigens due to various tumor-specific alterations. These alterations include genomic mutation, dysregulated RNA splicing, disordered post-translational modification, and integrated viral open reading frames [38,39]. A recent clinical trial explored the efficacy of an mRNA vaccine in combating pancreatic cancer [40]. These studies demonstrate the potential of bioinformatics tools in designing mRNA vaccines against various pathogens, including cancer.

## 5. Online Tools and Their Applications for Vaccine Design

This section provides a comparison of in silico tools applied to reverse vaccinology that have been utilized or have the potential to aid in the design of mRNA vaccines against SARS-CoV-2 [11] (Figure 2).

### 5.1. Vaccine Construction

#### 5.1.1. Retrieval of Sequence

The initial phase preceding vaccine development entails the acquisition of the viral sequence, which serves as the principal focus for immunization strategies. This can be accomplished by querying publicly accessible repositories replete with extensive data on viral genomic sequences, which are continually augmented with novel information. The National Center for Biotechnology Information (NCBI) emerges as a preeminent database, offering an exhaustive compendium of both genomic and proteomic sequences. The FASTA format is a text-based format used to represent amino acid sequences of the target protein. Notably, it is the most frequently utilized resource, with approximately 66.7% of the articles examined in this review relying on NCBI for sequence extraction and subsequent bioinformatic investigations.

The Virus Pathogen Database and Analysis Resource (ViPR) serves as another specialized repository with a focus on viral pathogens. Studies cited as Ahammad and Lira [27], ELKHOLY [28] leveraged ViPR for the purpose of sequence acquisition, whereas the Global Initiative for Sharing All Influenza Data (GISAID) was utilized in Oluwagbemi et al. [30], Oladipo et al. [32], Hussain et al. [33] for the procurement of complete genomic sequences of SARS-CoV-2 and its associated variants. GISAID is particularly renowned for its expeditious data dissemination capabilities, rendering it indispensable for research endeavors necessitating real-time genomic data. The study cited as Khan et al. [31] employed Uniprot, an exhaustive compendium of protein-related data. Additionally, the Protein Data Bank (PDB) was cited as the data source in studies Bhattacharya et al. [29], Durojaye et al. [34].

#### 5.1.2. T Cell Epitope Prediction

Generally, The Immune Epitope Database (IEDB) stands as a premier source of immunomic and host tools, offering crucial insights for epitope prediction and analysis [41]. Notably, tools like NetMHCpan (4.0) and NetCTL are distinguished for their foundation on artificial neural networks (ANNs) [42,43]. Furthermore, ProPred1 deserves mention as an online tool tailored for MHC I epitope prediction, incorporating matrices for 47 MHC Class I alleles, along with proteasomal and immunoproteasomal models [44]. Some studies indicate that using a combination of two servers enhances the reliability of the results [45,46].

Through computational algorithms designed to forecast cytotoxic T lymphocyte (CTL) epitopes, researchers can design vaccines that are more likely to elicit a strong immune response [22]. Among the reviewed works, eight cases studied these epitopes. The frequent use of NetCTL is confirmed by its application in the reviewed papers. Although IEDB was used three times, Ahammad and Lira [27] used it as a second layer of verification. It is common for researchers, after identifying potential epitopes using NetCTL, to turn to IEDB for a second layer of verification, thus ensuring the robustness and reliability of their predictions. NetMHC 4.1, ProPed1, Epijen, and RankPep were used less frequently, with each being utilized only in one of the reviewed studies.

Painter et al. [47] demonstrated that mRNA vaccines activate T cells specific to SARS-CoV-2, playing a pivotal role in ensuring long-lasting immunity. Notably, the swift priming of CD4+ T cells underscores their foundational role in preparing for a robust adaptive immune response, particularly following a booster dose. As detailed in Table 1, the IEDB software was exclusively employed for predicting MHC II-bound epitopes. This highlights the superior performance of machine learning models over other in silico prediction methods, as emphasized by [48]. Of the studies reviewed, six explored these epitopes. It is important to note that the percentile selection criterion might vary depending on the specific objectives of the study. For instance, Ahammad and Lira [27], Oluwagbemi et al. [30] focused on a percentile selection criterion of ≤0.25, while Bhattacharya et al. [29] used the default settings.

Despite the significant progress made, the accuracy of peptide–MHC II binding algorithms remains notably inferior to that of MHC class I binding predictors, underscoring the persistent challenges in the field. The MHC class II epitope prediction tools, particularly those from IEDB, are showing promise [49]. However, NETMHCIIpan-2.0 stands out as the best-performing method, with the capability to predict epitope binding with minimal or even absent experimental data [50].

#### 5.1.3. Continuous and Discontinuous B Cell Epitope Prediction

Several authors employed methodologies focused on the identification of linear epitopes. Ahammad and Lira [27] used the iBCE-EL server, while ELKHOLY [28] enhanced his research with more tools such as Rankpep. Bhattacharya et al. [29] utilized the BCPREDS server, while Oluwagbemi et al. [30], Oladipo et al. [32] combined BcePred with iBCE-EL and ABCpred server predictions, respectively. Similarly, Durojaye et al. [34] integrated ABCpred, iBCE-EL, and Bepipred web servers. For the prediction of discontinuous B cell epitopes, Hussain et al. [33] exploited the capabilities of DiscoTope v2. The multi-method approach, as shown by Pourseif et al. [35], represents an advancement in prediction strategies, using Bepipred for continuous and Discotope for discontinuous epitopes.

#### 5.1.4. Antigenicity

Vaxijen is a commonly used tool for predicting the antigenicity of T cell epitopes [51]. This software evaluates the antigenicity of a target organism, such as a virus, bacterium, tumor, parasite, or fungus. For this analysis, a threshold of ≥0.5 was used because most models had their highest accuracy at this threshold [11,52]. This tool was employed in Ahammad and Lira [27], ELKHOLY [28], Oluwagbemi et al. [30], Durojaye et al. [34], and Khan et al. [31] for epitope screening, considering only 55.6% of the total articles reviewed.

#### 5.1.5. Immunogenicity

The IEDB Analysis Resource is a commonly used tool for predicting the immunogenicity of CTL epitopes. It provides both negative and positive values, with positive values indicating potential immunogenicity. This tool was employed in Ahammad and Lira [27], ELKHOLY [28], and Oluwagbemi et al. [30] for initial epitope screening, considering only 33% of the total articles reviewed.

The CD4episcore tool provides a strong framework for predicting CD4+ T cell immunogenicity, with its effectiveness validated across various techniques for epitope identification, antigen sources, and ethnicities [53]. However, it is important to note that none of the studies reviewed addressed this aspect, highlighting the need for further validation of the predicted optimization of helper T lymphocyte (HTL) epitopes.

#### 5.1.6. Allergenicity

Allergenicity prediction is a critical step in therapeutics due to its involvement in predicting the cross-reactive potential of novel proteins from the sequence identities of known allergens [54].

The AllerTop 2.0 server is an online tool that can be used to predict the allergic or non-allergic nature of potential epitopes. The epitopes entered are evaluated individually, and the server provides a result indicating whether the sequence is likely to be allergenic or non-allergenic, along with a link to a protein with a similar sequence [45,55]. This tool was employed in Ahammad and Lira [27], ELKHOLY [28], Bhattacharya et al. [29], Oluwagbemi et al. [30], Durojaye et al. [34], and Khan et al. [31] for epitope screening, considering only 66.7% of the total articles reviewed.

#### 5.1.7. Toxicity

Toxicity profiles were assessed using ToxinPred, a specialized server that employs support vector machine (SVM) models for the classification of epitopes as either toxic or non-toxic. In this study, ToxinPred served as the exclusive tool for screening non-toxic epitopes. A review of nine related articles revealed that only Ahammad and Lira [27], ELKHOLY [28], and Oluwagbemi et al. [30] conducted this particular type of analysis [56].

#### 5.1.8. Inducibility of Interferon-γ (IFN-γ), Interleukin-4 (IL-4), and Interleukin-10 (IL-10)

In the quantitative analysis conducted in this study, it was found that approximately 33.3% of the reviewed papers focused on predicting the inducibility of interferon-γ (IFN-γ) using the IFNepitope server [57]. Similarly, 33.3% of the papers were devoted to the prediction of interleukin-4 (IL-4) inducibility using the IL4pred server [58], and 22.2% targeted interleukin-10 (IL-10) inducibility using the IL10pred server [59].

#### 5.1.9. Population Coverage

Population coverage indicates the proportion of the population that could potentially benefit from the vaccine. IEDB Population Coverage tool is used to predict the size of the population that would elicit an immune response to the constructed vaccine [60]. Among nine articles, only 55.6% performed this analysis.

### 5.2. Primary Vaccine Construct

In the realm of mRNA vaccine development, the design of the construct is paramount, influencing the vaccine’s efficacy, stability, and safety. Ahammad and Lira [27] and ELKHOLY [28] offer a comprehensive insight into this crucial phase, deepening our understanding of the entire developmental trajectory. The open reading frame (ORF) emerges as an essential component of the mRNA vaccine. It should encompass five pivotal elements: the Kozak sequence, epitopes, adjuvants, linkers, and a stop codon. The Kozak sequence, a consensus sequence vital for efficient mRNA translation, incorporates the start codon. Conversely, the surrounding sequence of the stop codon can be optimized to effectively terminate the translation. Chosen epitopes should exhibit antigenicity, be non-allergenic, non-toxic, and, exclusively for HTLs, induce cytokines. Adjuvants amplify the immune response, linkers seamlessly connect various construct segments, and the stop codons ensure translation termination. This synthesis draws from a range of articles, offering a detailed perspective on this foundational stage and enhancing our comprehension of the entire process [61].

On the other hand, Bhattacharya et al. [29] and Oladipo et al. [32] focus on the stabilizing elements like the 5${}^{\prime }$ cap and poly(A) tail, which are indispensable for mRNA stability and efficient translation. Notably, Bhattacharya et al. [29] introduces the innovative concept of self-amplifying mRNA vaccines, offering a mechanism for enhanced efficacy. Adjuvants also play a vital role in boosting the vaccine’s adaptive immune response. While ELKHOLY [28] and Oluwagbemi et al. [30] highlight the CD40 ligand (CD40L) as a co-stimulatory molecule, Khan et al. [31] introduces human beta defensin 2 (HbD-2) as another potential adjuvant. These molecules activate professional antigen-presenting cells (pAPCs), adding complexity and potency to the vaccine construct.

Ahammad and Lira [27], ELKHOLY [28], and Oluwagbemi et al. [30] discuss how specific linkers like GPGPG and (EAAK)2 are employed to optimize the vaccine’s efficacy. Hussain et al. [33] employs molecular modeling to assess the structural and thermodynamic attributes of the construct, a technique further refined by Durojaye et al. [34] through the use of SimRNA for 3D structure prediction. Pourseif et al. [35] add to the discussion by presenting two distinct vaccine constructs optimized for different platforms, showcasing the adaptability inherent in mRNA vaccine technology. The use of specific linkers and molecular modeling techniques can help optimize the efficacy of mRNA vaccines, while the adaptability of the technology allows for the creation of different vaccine constructs for different platforms [62].

### 5.3. Post-Vaccine Construction

#### 5.3.1. Allergenicity, Antigenicity, and Solubility Profile Analysis of the Vaccine Construct

The evaluation of allergenicity, antigenicity, and solubility profiles is indispensable for ensuring both the safety and efficacy of mRNA vaccine development. A variety of bioinformatics tools and servers are employed to scrutinize these critical properties, as highlighted by the articles under review and detailed in Table 2.

Antigenicity gauges the ability of a vaccine to trigger an immune response. Ahammad and Lira [27], Bhattacharya et al. [29], Oluwagbemi et al. [30], Pourseif et al. [35], and Oladipo et al. [32] utilized the VaxiJen server for this purpose, with Ahammad and Lira [27] and Oluwagbemi et al. [30] also incorporating the ANTIGENpro server that employs machine learning algorithms. These methods are generally alignment-free and rely on the physicochemical characteristics of the protein to make their predictions.

Allergenicity is another vital aspect that needs meticulous evaluation to ensure the vaccine’s safe administration. The AllerTOP server was commonly used for this purpose [27,29,30,32,35]. Oluwagbemi et al. [30] extended its analysis by also using the AllergenFP server, while Pourseif et al. [35] included FAO/WHO allergenicity rules in its evaluation. These tools employ various methods like autocross-covariance (ACC) transformation and E-descriptors to assess the likelihood of an allergic response. The toxicity of the vaccine construct is a critical safety parameter. Ahammad and Lira [27], Oluwagbemi et al. [30], and Oladipo et al. [32] used the ToxinPred server for this assessment. This server operated based on support vector machine (SVM) models, aiding in the classification of toxicity and non-toxicity [56].

The solubility of the expressed vaccine protein is not to be overlooked, as indicated by Bhattacharya et al. [29], who employed the Protein-Sol online server for this assessment. Solubility is crucial for a vaccine’s stability and its subsequent effectiveness in eliciting an immune response. Physicochemical properties such as molecular weight, theoretical isoelectric point (pI), and instability index (II) are also integral to vaccine construct evaluation. Ahammad and Lira [27], Bhattacharya et al. [29], Oluwagbemi et al. [30], and Pourseif et al. [35] utilized the ExPASy ProtParam online web server for this comprehensive analysis. Pourseif et al. [35] went a step further by also examining additional properties like half-life and extinction coefficient.

#### 5.3.2. Secondary Structure Vaccine

Various bioinformatics tools have been developed to assist in the process of secondary structure configuration, each with its own features, and selected based on the type of work and purpose. Therefore, 44.4% of the papers reviewed addressed this analysis. PSIPRED v4.0 and SOPMA applications were used by Pourseif et al. [35] and Oluwagbemi et al. [30], respectively [63,64]. Using default parameters, these two servers calculate the percentage of 2D configurations such as alpha helix, random coil, and beta-turn. Moreover, specific servers for predicting RNA sequence structures, such as RNAfold and simRNA, were used by Hussain et al. [33] and Durojaye et al. [34], respectively [65,66]. The latter was proposed to predict the three-dimensional structure, but also provides additional results, including the secondary structure.

#### 5.3.3. Tertiary Structure Vaccine

Several tools are used to assist in the configuration of tertiary structure. According to quantitative investigations in this study, approximately 90% of the papers have been devoted to this analysis. SWISS-MODEL is the most frequently used, appearing in approximately 44.4% of the studies. The tool’s popularity is largely due to its homology modeling capabilities [67]. Following SWISS-MODEL is Phyre2, which is used in about 22.2% of the studies. Phyre2 is particularly renowned for its ability to generate high-quality 3D models even when sequence identity to known structures is low [68]. Less commonly employed are SimRNA and Robetta, each appearing in 11.1% of the studies. SimRNA specializes in RNA structure prediction, while Robetta is distinguished for its ab initio modeling capabilities [65,69].

#### 5.3.4. Molecular Docking

Molecular docking is a critical step in vaccine development, providing insights into the binding affinity and interactions between epitopes and their corresponding major histocompatibility complex (MHC) alleles or other TLRs. Each article in the review offers a unique approach to molecular docking, employing various algorithms and software tools.

Ahammad and Lira [27] performed a comprehensive docking analysis between T lymphocyte epitopes and their corresponding MHC alleles. The study used AutoDock Vina for docking and evaluated the binding affinity in terms of kcal/mol. Bhattacharya et al. [29] also focused on molecular docking between epitopes and MHC alleles, but extended the study to include docking with TLR7. The study employed the HDOCK server for protein docking against mRNA molecules. Both Oluwagbemi et al. [30] and Pourseif et al. [35] utilized the ClusPro 2.0 server to assess the binding affinity between their vaccine constructs and TLRs. While Oluwagbemi et al. [30] focused on interactions with TLR4 to gauge the vaccine’s potential efficacy in immune signaling pathways, Pourseif et al. [35] extended the analysis to include TLR5 as well. This additional consideration of multiple TLRs by Pourseif et al. [35] adds depth to our understanding of how the vaccine might elicit a broad immune response.

Khan et al. [31], Durojaye et al. [34], and Oladipo et al. [32] employed various computational tools for peptide docking and molecular dynamics simulations to understand the vaccine’s interaction with different components of the immune system. Durojaye et al. [34] used both HPEPDOCK and ClusPro servers to dock peptides against the HLA*A-0201 allele, and further validated the stability of the protein–peptide complex through molecular dynamics simulations. On the other hand, Khan et al. [31] utilized the HawkDock server for docking analysis with human TLR4, and also examined the interactions of selected T cell epitopes with corresponding HLAs. Meanwhile, Oladipo et al. [32] leveraged Hex 8.0.0 software to study the vaccine construct’s affinity for TLR3 and TLR9, which are known for recognizing microbial or viral nucleic acids. This diverse set of methodologies offers a comprehensive understanding of how the vaccine may engage with and activate various immune system components.

#### 5.3.5. Molecular Dynamics and Quantum Calculations

According to quantitative investigations in this study, approximately 44.4% of the papers have been devoted to molecular dynamics analyses. It is noteworthy that a majority of the articles did not employ molecular dynamics simulations in their research, which raises questions about the comprehensiveness of their computational analyses.

The molecular dynamics simulation of complex vaccine–receptor interactions may be an alternative and complementary tool for clarifying the physical basis of the structure and functions of biomolecules, mainly due to the prominence that mRNA vaccines have been gaining lately [62,70,71]. The study by Oluwagbemi et al. [30] stood out for its application of MD simulations, employing the iMODS server and the WebGro macromolecular simulations platform. The analysis included a detailed evaluation of the vaccine-TLR4 docked complex, utilizing normal mode analysis (NMA) and various equilibrium properties like RMSD and RMSF.

Durojaye et al. [34] also employed MD simulations but focused on the design of the epitope peptide and its interactions with the HLA*A-0201 receptor. The GROMACS and D3Pockets servers were used to model, while Pourseif et al. [35] used MD simulations using only the GROMACS 5.0.7 software and the GROMOS 96 force field to optimize the free energy of the model. The article by Oladipo et al. [32], on the other hand, used the IMODs tool to capture the detailed atomic behavior of the proteins and their interactions with the TLR3 and TLR9 complexes.

MD simulations provide a dynamic, time-dependent perspective on molecular interactions and changes, while NMA offers a static view focused on inherent vibrational modes [70,72]. Using both methods together allows for a more robust and comprehensive analysis, validating findings and thereby enhancing the robustness and comprehensiveness of their analyses [30].

While this is a valuable approach, the studies could benefit from the incorporation of additional computational techniques. For instance, employing quantum mechanical calculations could provide a more nuanced understanding of electronic interactions within the complexes. To enhance the accuracy of calculations involving intricate vaccine–receptor interactions, a hybrid approach that combines quantum mechanics/molecular mechanics (QM/MM) can be employed. QM/MM techniques have firmly established themselves as the pinnacle of computational methods for biomolecular systems by this point. The burgeoning number of publications that have leveraged QM/MM methodologies serve as compelling evidence of their maturation since their inception approximately three decades ago [11,45,73,74].

The QM/MM methodology enables the decomposition of the total energy into distinct components, thereby facilitating a nuanced analysis of the protein environment, down to individual residues. This is particularly advantageous when numerous electrostatic interactions are present [75]. In a hypothetical application of this methodology using the ONIOM multilayer framework, accessible in Gaussian code, the receptor would be designated to the MM layer, while key amino acid residues from the vaccine would be assigned to the QM layer. The B3LYP (Becke, three parameters, Lee–Yang–Parr) hybrid functional and the 6-311G (d,p) basis set would be used for the QM calculations. Amino acid residues within a 6.0 Å radius from the ligand’s centroid would be allowed to undergo geometric optimization [11,45,46].

#### 5.3.6. Computational Immune Simulation Analysis of the Constructed Vaccine

The utilization of in silico immune simulations in vaccine development has become an increasing tool for predicting real-world outcomes. The studies under review employ a variety of approaches to this end, revealing both the potential and the limitations of current methodologies. At this stage, the evaluation of the immunogenicity of all the predicted conjugate vaccine peptides and the characteristics of the immune response is carried out through in silico immune simulation. Notably, 77.8% of the reviewed articles evaluated the immune response profile, underscoring the importance of this aspect in vaccine research. The C-ImmSim online server predicts the associated immune interactions and epitopes using a machine learning-based method and position-specific scoring matrix (PSSM) [76].

Ahammad and Lira [27], Hussain et al. [33], and Bhattacharya et al. [29] appear to be almost identical in their approach, employing the C-ImmSim server with default settings for predicting epitopes and immune interactions. However, Bhattacharya et al. [29] adds a layer of complexity by incorporating HLA alleles for immune profiling, offering a more personalized prediction of vaccine efficacy. The use of default settings may be convenient but potentially overlooks the nuances of individual immune responses, a concern that the authors of [32] addressed by customizing the total number of time steps in the simulation. This customization allows for a more tailored approach, potentially offering insights that are closer to real-world scenarios.

The work of Oluwagbemi et al. [30] is distinguished by its detailed simulation of anatomical compartments, providing a more holistic understanding of vaccine–immune system interactions. Khan et al. [31] also stand out for their customization of the total number of time steps, allowing for a nuanced understanding of the immune response over time. Oladipo et al. [32], while also using C-ImmSim, emphasize the importance of recognizing antigenic peptides but stick to default settings, limiting the study’s depth.

Notably, ELKHOLY [28], Durojaye et al. [34], and Pourseif et al. [35] did not employ any in silico immune simulations, representing a significant gap in these studies. The absence of such simulations limits the predictive power of these studies and raises questions about the robustness of their conclusions.

#### 5.3.7. In Silico Codon Optimization and Molecular Cloning

The in silico codon optimization and molecular cloning stage is a critical component in the development of mRNA vaccines, as it directly impacts the translation efficiency and expression of the vaccine construct. However, among the nine articles, only Bhattacharya et al. [29] and Khan et al. [31] delve into this aspect, employing different tools and methodologies.

Bhattacharya et al. [29] use the JCat server for codon optimization based on default parameters, which are generally considered crucial for successful recombinant DNA cloning. The study goes a step further by simulating the ligation of the vaccine candidate into an *E. coli* K12 expression vector, using the EMBOSS Backtranseq tool and SnapGene software. Khan et al. [31] also employ the JCat tool, but complement it with the ExpOptimizer tool. While JCat focuses on codon adaptation based on a proposed algorithm stored in the PRODORIC database, ExpOptimizer aims to highly express any protein of interest in any mainstream expression host.

Notably, the other articles do not address this stage, representing a significant gap in their methodologies. The absence of in silico codon optimization and molecular cloning simulations could limit the applicability and translational success of these studies, raising questions about the completeness of their vaccine development process. This approach optimizes the codon sequence and provides a simulated environment for assessing the cloning efficiency, thereby enhancing the study’s robustness.

## 6. Conclusions

The integration of reverse vaccinology to design mRNA vaccines plays a pivotal role. The blend of computational tools with traditional methods is catalyzing the swift identification and assessment of crucial epitopes, streamlining the pathway to robust vaccine designs. This comprehensive approach ensures not only the enhanced stability and efficiency of mRNA vaccines, but also broadens the scope of their applicability against diverse pathogens, including the emerging variants of SARS-CoV-2.

Many studies have employed a reverse vaccinology approach for COVID-19, targeting the SARS-CoV-2 spike glycoprotein. These studies span a broad spectrum, from epitope binding predictions to molecular dynamics and immune simulations Ahammad and Lira [27], ELKHOLY [28], Bhattacharya et al. [29], Oluwagbemi et al. [30], Khan et al. [31], Oladipo et al. [32]. To address the emerging SARS-CoV-2 variants, the DOW-21 mRNA vaccine construct was developed, highlighting crucial spike protein domains and fine-tuned for efficient translation Hussain et al. [33]. Another study harnessed reverse vaccinology for an mRNA vaccine, concentrating on the peptide sequence “YLQPRTFLL” and customizing it for the human toll-like receptor 7 Durojaye et al. [34]. Furthermore, innovative approaches led to the exploration of two mRNA vaccine blueprints, namely, DPVC and SAMV. In silico evaluations affirmed their potency Pourseif et al. [35], with the research also examining elements influencing vaccine compatibility and resilience through molecular dynamics simulations.

However, the incorporation of advanced computational techniques, such as molecular dynamics (MD) simulations and quantum mechanics/molecular mechanics (QM/MM) methods, could offer more insights into vaccine–receptor interactions and enhance the robustness of these in silico models [11,45,73,74]. The absence of such advanced computational techniques leaves room for further refinement in future research. As we transition from identifying potential areas of improvement to offering solutions, it is clear that the landscape of mRNA vaccine design is evolving.

We hope this review acts as a valuable guide for researchers and developers keen on leveraging reverse vaccinology for mRNA vaccine design against SARS-CoV-2 and other emerging pathogens. The adoption of reverse vaccinology in mRNA vaccine creation represents not only a solution to today’s global health issues, but also lays the groundwork for a new era in vaccine development.

## Figures and Tables

**Figure 2 viruses-15-02130-f002:**
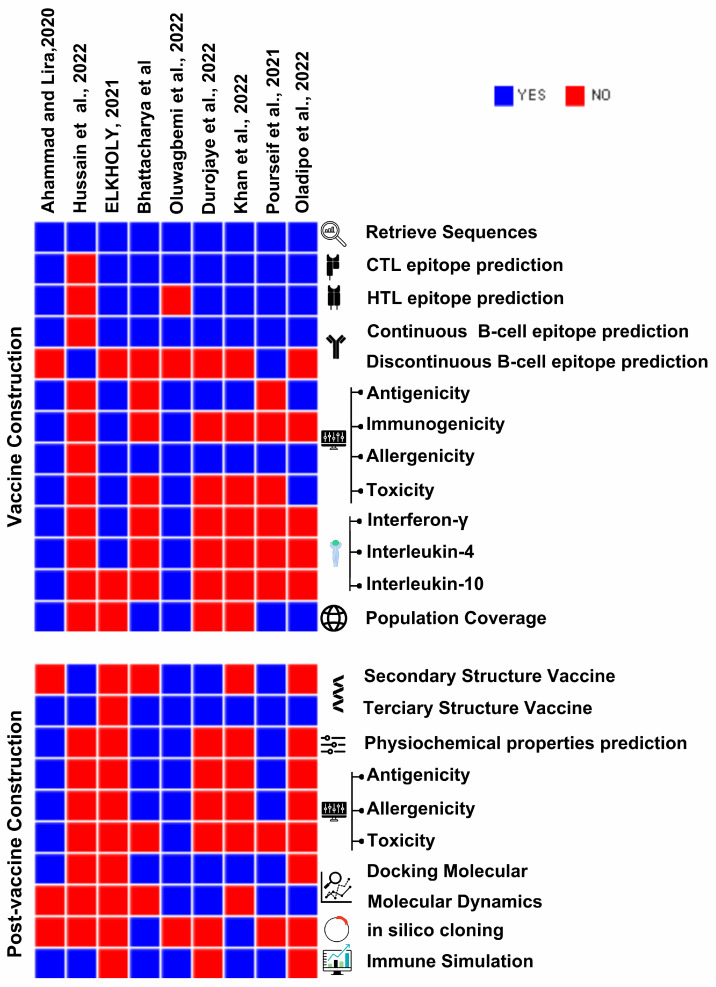
Heat maps for the vaccine construction and post-vaccine construction phases, the values of which correspond to the reviewed articles by mRNA steps from each phase of mRNA vaccine design [27,28,29,30,31,32,33,34,35]. The bar chart shows which articles carried out each step, where blue corresponds to “YES” and red corresponds to “NO”.

**Table 1 viruses-15-02130-t001:** Comparison of various in silico tools used in vaccine construction studies. The table shows the frequency of each tool’s usage in the revised articles.

Vaccine Construction			
	In Silico Software	f${}_{i}$/N × 100 ${}^{*}$	Reference
**Retrieve Sequences**	Genbank	55.6% (159)	[27,28,29,32,35]
	Gisaid	33.3% (3/9)	[30,32,33]
	ViPR	22.2% (2/9)	[27,28]
	NCBI	77.8% (7/9)	[28,29,30,33,34,35]
	Uniprot	11.1% (1/9)	[31]
	PDB	22.2% (2/9)	[29,34]
**CTL epitope prediction**	NetCTL	66.7% (6/9)	[27,28,29,30,31,32]
	IEDB	33.3% (3/9)	[27,34,35]
	NetMHC 4.1	22.2% (2/9)	[28,34]
	ProPed1	11.1% (1/9)	[34]
	EpiJen	11.1% (1/9)	[34]
	Rankpep	11.1% (1/9)	[28]
**HTL epitopes prediction**	IEDB	77.8% (7/9)	[27,28,29,30,31,32,35]
**Continuous and Discontinuous B cell epitope prediction**	ABCpred	55.6% (5/9)	[28,31,32,34,35]
	BCpred	33.3% (3/9)	[29,30,32]
	iBCE-EL	22.2% (2/9)	[27,30]
	BcePred	33.3% (3/9)	[29,30,34]
	Bepipred	22.2% (2/9)	[34,35]
	Discotope	22.2% (2/9)	[33,35]
**Antigenicity**	VaxiJen 2.0	55.6% (5/9)	[27,28,30,31,34]
**Immunogenicity**	IEDB (CTL)	33.3% (3/9)	[27,28,30]
**Allergenicity**	AllerTop V 2.0	66.7% (6/9)	[27,28,29,30,31,34]
**Toxicity**	Toxinpred	33.3% (3/9)	[27,28,30]
**Interferon-γ (IFN-γ)**	IFNepitope	33.3% (3/9)	[27,28,30]
**Interleukin-4 (IL-4)**	IL4pred	33.3% (3/9)	[27,28,30]
**Interleukin-10 (IL-10)**	IL10pred	22.2% (2/9)	[27,30]
**Population coverage**	IEDB	55.6% (5/9)	[27,29,30,32,35]

* In the third column, the percentages of the regularity of each web tool in articles are shown.

**Table 2 viruses-15-02130-t002:** Comparison of various in silico tools used in post-vaccine construction studies. The table shows the frequency of each tool’s usage in the revised articles.

Post-Vaccine Construction			
	In Silico Software	f${}_{i}$/N × 100 ${}^{*}$	Reference
**Secondary Structure Vaccine**	PSIPRED	11.1% (1/9)	[35]
	RNAfold	11.1% (1/9)	[33]
	SOPMA	11.1% (1/9)	[30]
	SimRNA	11.1% (1/9)	[34]
**Tertiary Structure Vaccine**	SWISS-MODEL	44.4% (4/9)	[27,29,33,35]
	Phyre2	22.2% (2/9)	[30,32]
	SimRNA	11.1% (1/9)	[34]
	Robetta	11.1% (1/9)	[31]
**Physiochemical properties prediction**	ProtParam	44.4% (4/9)	[27,29,30,35]
	Protein-Sol	11.1% (1/9)	[29]
**Antigenicity**	VaxiJen 2.0	44.4% (4/9)	[27,29,30,35]
	ANTIGENpro	22.2% (2/9)	[27,30]
**Allergenicity**	AllerTop V 2.0	22.2% (2/9)	[27,29,30,35]
	AlgPred	11.1% (1/9)	[35]
	AllergenFP	11.1% (1/9)	[30]
**Toxicity**	Toxinpred	22.2% (2/9)	[27,30]
**Docking Molecular**	AutoDock Vina	22.2% (2/9)	[27,29]
	HDOCK	22.2% (2/9)	[29,34]
	ClusPro 2.0	33.3% (3/9)	[30,34,35]
	HPEPDOCK	11.1% (1/9)	[34]
	HawkDock	11.1% (1/9)	[31]
**Molecular Dynamics**	iMODS	22.2% (2/9)	[30,32]
	WebGro	11.1% (1/9)	[30]
	GROMACS 5.0.7	33.3% (3/9)	[30,34,35]
	D3Pockets	11.1% (1/9)	[34]
**In silico Cloning**	Jcat	22.2% (2/9)	[29,31]
**Immune simulation**	C-ImmSim	66.7% (6/9)	[27,29,30,31,33,35]

* In the third column, the percentages of the regularity of each web tool in articles are shown.

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
