# Peer review of "Advances of Reverse Vaccinology for mRNA Vaccine Design against SARS-CoV-2: A Review of Methods and Tools"

_viruses, 2023, doi:10.3390/v15102130_

Round 1
Reviewer 1 Report
This article effectively summarizes reverse vaccinology's potential in designing mRNA vaccines against SARS-CoV-2 and offers a glimpse into its broader application against emerging pathogens, making it informative and engaging. They introduce the mRNA design by comparing the in-silicon tool. Here are several points the author should consider:
1, For Figure 1, it is better to depict the mechanism of mRNA vaccine by your understating, not just adjust other's figure.
2, Figure 2 can use a better present way, such as using the bar chart to show the percentage usage of the different keywords.
3, In conclusion, the author holds that the study by Oluwagbemi et al. serves as a benchmark. It's better to discuss the drawbacks of this method which will make the conclusion more conceivable.
Author Response
We would like to express our heartfelt gratitude for the time and dedication you have invested in reviewing our manuscript. Your insightful feedback and constructive comments have been of paramount importance in refining and enhancing the quality of our work. We truly appreciate your expertise and the care with which you approached our manuscript, ensuring its improvement.
Thank you once again for your invaluable contribution to our research.
Warm regards,
Prof. Dr. Jonas I. N. Oliveira
on behalf of the authors

Reviewer 2 Report
This review is comprehensive, so I have two minor suggestions.
1. The data of COVID-19 information in the second sentence ", infecting over 800 million people and 16 killing over 14 million by August 2023" in the introduction needs correction according to WHO COVID-19 dashboard.
2. In addtion to current discussion, the concerns and the potential adverse effect of mRNA vaccine should be discussed.
Author Response

(The authors gave the same response as above.)

Reviewer 3 Report
For the review article entitled “Advances of Reverse Vaccinology for mRNA Vaccine Design against SARS-CoV-2: A Review of Methods and Tools”, the authors have provided a nice overview of several different studies that have used computational methods for predicting mRNA vaccine efficiency in multiple different areas. The manuscript is well-written and reasonably well-structured. However, there are some major concerns. Assessment of the overall validity of reverse vaccinology (RV) is missing. How has RV been tested in practice and how close to the prediction model was it? It’s great that the authors list all these predictive tools for different aspects of sequenced-based vaccine development but, to this reviewer, the element of ‘how does it actually work?’ is missing. Furthermore, while there is extensive review on what tools each listed article used, I am lacking the overall point of the tool in the text. So what X et al. and Y et al. used this tool? How good is the tool? How does the tool work? Is it based on real data? Is it based on a machine learning approach? For part 2 “overview of reverse vaccinology in mRNA vaccine design”, the authors state, “This section provides a comparison of in silico tools applied to reverse vaccinology that have been utilized or have the potential to aid in the design of mRNA vaccines against SARS-CoV-2”, and yet, there is less of a review on the tools and more of just statements as to who used these tools. What is more important, is how well these tools work (perhaps based on any real life validation). What is better for predicting epitopes, IEDB? NetCTL? NetMHC 4.1? What does each tool offer (advantages and disadvantages)? Is it better to use two different tools to narrow in on protective epitopes? A very important section would be, based on all of these tools, how well do they predict current COVID-19 mRNA vaccinations?
For the “conclusions” section, the authors have not really shown any reason that RV is “a promising approach to design mRNA vaccines against SARS-CoV-2”. The direction that the current review has taken hasn’t really shown this. This just relates to the predictive efficacy of the tools (i.e. how well validated are they?).
As a positive, I will say that Tables 1 and 2, as well as Figure 3 are very nice. Figure 1 is also quite nice but are the authors sure that the mRNA coding sequence doesn’t contain the whole spike protein gene? To my knowledge, the mRNA vaccines encoded the whole spike protein. It doesn’t make logical sense to only include epitopes, as the protein conformation is highly relevant for epitope formation, especially for B cell responses. The authors should consider revising Figure 1B in the context of SARS-CoV-2. I don’t see the point of Figure 2? I would recommend deleting it.
I do believe that this review article is of relevance and contains some highly relevant information. As an approach, RV can be a super useful tool and should definitely be considered in future vaccine designs. The tools listed are also of great help to many readers. Therefore, I do believe that once the manuscript has the focus orientated correctly, it will be a great read for many.
Other things to note:
Introduction
“SARS-CoV-2” and “COVID-19” are abbreviations. Please provide the unabbreviated version with the abbreviated version in the parentheses first.
“Reverse vaccinology (RV) is a novel approach” and yet “RV emerged in the late 1990s”. So, the approach emerged over 20 years old and still considered novel? Here, the authors also state, “that uses the genomic information” and yet in the next section, “Reverse vaccinology is an approach that uses computational tools to analyze the proteome”. So, is it genome or proteome? I realize they are not mutually exclusive but if it’s based on the sequence, then it’s the genome. There needs to be consistency.
“Reverse vaccinology became particularly relevant for the development of vaccines against SARS-CoV-2, as it facilitated the rapid identification of the spike protein (S) as the key antigen responsible for the entry of the virus into cells and for the elicitation of neutralizing antibodies.”, I’m not sure if RV was really employed to identify the spike protein for the COVID-19 vaccines. It is more likely people knew that the spike protein is the main entry protein from which antibodies can be developed against (based on prior coronavirus research). Logically, it makes the most sense to use it as a vaccine target. I highly doubt that it was RV that facilitated this. Of course, the sequence of the spike protein was needed, but RV didn’t determine that this sequenced be used over others.
“Since then, reverse vaccinology has been used to identify antigen candidates for various pathogens, such as hepatitis C virus, influenza, and Zika”, identify… sure. But how well did they do in clinical trials, if taken to clinical trials? This is important.
2. Overview of reverse vaccinology in mRNA vaccine design
For the first paragraph (lines 66-74), can the authors please describe RV in more detail. The introduction to this whole approach is too vague. Is this a machine learning approach based on real data? I.e. how does the tool predict “protective responses” exactly? On that note, how does it then predict non-protective responses. When the authors say, “precise regions” are they referring to T cell epitopes more specifically? B cell epitopes are harder to predict as one can make an antibody response to almost anything. I’m guessing the precise regions are more protective?
Prior to the sequences ending up on NCBI, the virus needs to first be sequenced, generally via NGS. I think this step is essential before just looking up sequences on NCBI? Especially if it’s a novel pathogen.
On line 101, please remove “the” from “the cytotoxic T lymphocytes”. The definite form here is incorrect.
Please unabbreviate IEDB.
“Ahammad and Lira [15], ELKHOLY [16] lay the foundational framework by emphasizing the necessity of five key elements in the Open Reading Frame (ORF): the Kozak sequence, epitopes, adjuvants, linkers, and a stop codon. These elements serve as the building blocks of the mRNA vaccine construct”, can the authors please go into these 5 elements in more detail. For example, what is a Kozak sequence and why is it important? This will be important for readers.
“self-amplifying mRNA vaccines”, self-amplifying how?
“GPGPG and (EAAK)2 are employed to optimize the vaccine’s efficacy.”, provide a one sentence summary as to how these enhance efficacy.
The immunology parts of this whole section can be removed. The authors don’t need to stress the importance of B and T cells and should, instead, focus more on the tools. For example, “Upon recognition of specific epitopes, the cytotoxic T lymphocytes (CTLs) initiate the elimination of infected cells through the release of cytotoxic granules [24]. This highlights the important role that CD8+ T cells play in enhancing both cellular and humoral immune responses following vaccination”, the authors have not, in nearly enough detail, explained how “this highlights the important role”, especially for the humoral immunity. The lack of immunological understanding is evident and that is why I would recommend removing anything trying to explain the role of these lymphocytes. Instead, merely state that they are important with relevant citations and then focus on the predictive tools. Another example is, “The prediction of B-cell epitopes is essential for understanding how B-cells recognize antigens, present them to T-cells via MHC II, and ultimately generate an effective immune response through antibody production and immune memory formation”, the prediction of B cell epitopes is not “essential”. Generating real data on protective B cell responses is essential. Prediction can help but is not essential. Furthermore, B cells can develop in a CD4-independent manner, albeit to a less functional state. So, is this statement referring to B cell epitopes (i.e. antibody epitopes) or MHC II epitopes that stimulate B cell responses? Again, this vague explanation does not help the article. Focus on the tools.
“The utilization of in silico immune simulations in vaccine development has become an increasing tool for predicting real-world outcomes”, based on what current approved vaccines?
Conclusion
“Several studies have applied reverse vaccinology to design mRNA vaccines against SARS-CoV-2, and some of them have entered clinical trials.”, and how are the clinical trails going compared to the current approved mRNA vaccines? Better? Worse? How have the predictive models helped and how do they compare?
“Notably, our analysis reveals that the study by Oluwagbemi et al. [18], stands out for its comprehensive approach, covering a wide array of analyses from epitope binding prediction to Molecular Dynamics and Immune Simulation. This study serves as a benchmark for how to employ reverse vaccinology effectively in the design of mRNA vaccines.”, so the conclusion is, more analyses means a better outcome? How was this validated and why would this serve as a benchmark?
Author Response

(The authors gave the same response as above.)

Round 2
Reviewer 3 Report
The manuscript by da Silva et al. shows clear signs of improvement. The structure is now much easier to follow and understand. It’s great to see there is much more direction/story telling to the review. The messages are clear enough and, more importantly, the authors provide enough details about each bioinformatic tool, which will be extremely useful to readers. The development of mRNA vaccines has become increasingly prominent, and this review will help guide those looking to develop mRNA technologies, including myself.
There are some minor typing or grammatical errors detected, which I have listed below. Once fixed, I believe the manuscript is ready for publication.
Line 41, “region of genome, usually a protein” should be “region of genome, usually encoding a protein”. And, add “that” before “, can”.
Line 56, missing a space between “thatemploys”.
Line 82, should be “encodes”.
Line 93, should be “serves”.
Line 98, should be “comes”.
Line 200, CTL needs to be unabbreviated.
Line 253, HTL needs to be unabbreviated.
On line 286, “ORF” is unabbreviated here, but the term ‘ORF’ is used previously in the text. Just need to move the unabbreviated version up.
On line 307, remove “the” for “the hussain et al.”.
TLR has been unabbreviated on many occasions. You just need to do it once (line 144)
Line 487, should be “do not address”
Minor typing or grammatical errors detected.
Author Response
Dear Reviewer,
We would like to express our gratitude for your thoughtful and constructive feedback on our manuscript. Your keen observations and positive remarks are truly invaluable to us. We are particularly pleased to hear that the revisions have made the manuscript easier to follow and understand. It is reviewers like you who make it possible for us to improve and refine our work. Your insights will not only benefit us but also those looking to develop mRNA technologies, including yourself, as you kindly mentioned.
Attached is a .docx file containing all the corrections made in response to your comments.
Sincerely yours,
Prof. Dr. Jonas I. N. Oliveira
on behalf of the authors
